# An empirical study of drone medical logistics transportation in a multi-campus model of Chinese public hospitals: Real-world data-driven validation of timeliness and application effects

Xingbo Long[1]*, Min Huang[2], Xiankai Xie[1], Yanxin Wang[1], Xiaojiang Yu[1], Wei Zhou[1], Bo Hu[1]

**1** Department of Medical and Information Engineering, Deyang People's Hospital, Deyang, Sichuan Province, China, **2** Department of Quality Improvement, Deyang People's Hospital, Deyang, Sichuan Province, China

* 723766496@qq.com

## Abstract

In the real-world application scenario of China's public hospital multi-site operation model, this study systematically evaluates the feasibility, timeliness, and stability of drone-based medical logistics transportation between hospital campuses, providing high-quality empirical evidence for optimizing cross-campus medical supply transfer processes. This retrospective analysis collected complete drone medical logistics transport data between two campuses of Deyang People's Hospital from April 8 to July 30, 2024. The primary outcome measure was drone transport time per unit distance (min/km), compared with road traffic time per unit distance measured by three mainstream navigation apps (Baidu Maps, Amap, and Tencent Map) at different time points. Intergroup differences were analyzed using the Mann-Whitney U nonparametric test, with effect sizes calculated via Cliff's Delta. Statistical significance was set at $P < 0.05$. A total of 750 valid drone flight records were included, covering a distance of $(5.95 \pm 0.03)$ km with a unit time of $(1.64 \pm 0.14)$ min/km. The unit time for drone transportation was significantly lower than that measured by Baidu Maps $(2.06 \pm 0.12$ min/km), Amap $(2.01 \pm 0.12$ min/km), and Tencent Map $(2.03 \pm 0.09$ min/km) at the 0-point road traffic unit time (all $P < 0.001$). At all nine time points monitored by Tencent Map, the unit time per kilometer exceeded that of the UAV. During the 10:00 peak period, Tencent Map recorded a unit time of $(4.12 \pm 0.09)$ min/km, with the UAV achieving a time savings rate of 60.2%. Mann-Whitney U tests revealed significant differences across all time points $(P < 0.001)$, with Cliff's Delta absolute values consistently exceeding 0.75, indicating extremely large effect sizes. Drone-based medical logistics demonstrated significant advantages in timeliness and stability under the multi-campus model of urban hospitals, particularly during peak traffic congestion periods. This study provides crucial empirical support for establishing an efficient, intelligent

**Data availability statement:** All relevant data are within the paper and its Supporting Information files.

**Funding:** This work was supported by National Institute of Hospital Administration [grant number 2025RWE023]; Sichuan Province Health Information Center [grant number 2021ZXKY06012]; Sichuan Province Health Information Society [grant number 2022051]; and Sichuan Provincial Hospital Association [grant number SCZB013].

**Competing interests:** The authors have declared that no competing interests exist.

medical logistics system, holding significant implications for enhancing healthcare service efficiency and improving public health emergency response capabilities.

---

## 1. Introduction

### 1.1. Research background and significance

With the advancement of China's healthcare reform, public medical resources have become insufficient to meet the growing demand for multi-tiered medical services. Concurrently, urban expansion and new district development have driven large-scale public hospitals to actively implement the "one hospital, multiple campuses" model [1,2]. This strategy involves establishing new facilities in emerging urban zones to expand high-quality medical resources and alleviate service pressures. During the COVID-19 pandemic in 2020, public hospitals demonstrated critical importance in health emergencies, prompting a policy shift from strict scale control to proactive multi-campus development. In June 2021, the State Council General Office's Opinions on Promoting High-Quality Development of Public Hospitals [2] explicitly stated support for certain strong public hospitals to moderately develop multiple campuses while controlling the scale of individual facilities. This policy direction has significantly driven the spatial restructuring of the healthcare service system. Currently, many public hospitals in China, particularly tertiary hospitals, have adopted a multi-campus development model, giving rise to an integrated management approach for hospitals with multiple locations.

While multi-campus operations enhance service accessibility [3], they introduce management challenges, including cost-control complexities and inter-campus logistics coordination. Resource limitations in branch campuses necessitate specimen transfers for various pathological examinations [4], creating operational bottlenecks. Current medical logistics in Chinese public hospitals predominantly rely on manual transportation via scheduled ambulance shuttles [5]. This system exhibits three principal limitations: prolonged response times, inconsistent efficiency, and absence of real-time tracking mechanisms, compromising specimen security and timeliness. These deficiencies particularly affect acute care transfers and intraoperative frozen pathology deliveries [6], where stringent time requirements render traditional methods inadequate.

Against this backdrop, drone technology emerges as an innovative solution to multi-campus medical logistics challenges. As a novel logistics approach, it offers distinct advantages: freedom from ground traffic constraints, rapid response capabilities, and high cost-effectiveness.

### 1.2. Literature review

Research on the application of drones in the medical field has developed rapidly in recent years, primarily focusing on the following directions:

**1.2.1. Types and application scenarios of medical drone transportation.** The application of medical drones has progressed from theoretical exploration to

multi-scenario validation. Medical drone transportation can be categorized into four major types:(1) Diagnostic sample transportation (including blood, urine, tissue specimens, etc.) [7];(2) Pharmaceutical transportation (routine medications, emergency drugs, vaccines, etc.) [8]; (3) Medical equipment transport (small medical devices, emergency equipment, etc.) [9]; (4) Emergency medical response (e.g., urgent blood transport) [9–11]. Among these, diagnostic sample transport has become the most maturely researched area in drone medical applications due to its high time-sensitivity requirements and the samples' small size and light weight.

Studies indicate that drones can maintain the integrity of biological samples during transport. Stierlin et al. [7,12,13] compared the quality metrics of blood samples transported by drones versus ground transportation and found no significant differences in results for complete blood counts, biochemical tests, or coagulation function tests. Similarly, Stierlin et al. [14,15] confirmed that the analytical results of urine samples transported by drones showed high consistency with those transported by traditional methods. These studies lay the scientific foundation for the application of drones in medical sample transportation.

**1.2.2. Limitations of existing research and innovations of this study.** Despite progress in drone medical transport research, the following limitations remain: (1) Most studies rely on simulated environments or small-scale trials, lacking real-world large-scale application data [9, 16–21]; (2) Evaluations of transport timeliness often focus on single time points or short cycles, failing to comprehensively reflect performance under varying traffic conditions [22]; (3) Research addressing the specific scenario of multi-campus urban hospitals in China remains scarce. An Israeli case study [23] explored the application of vertical takeoff and landing (VTOL) drones within an urban medical network, offering direct reference value for optimizing internal logistics within large medical groups in Chinese cities. However, similar studies have not been conducted within the Chinese context. (4) Systematic research on the impact of environmental factors (e.g., weather) on drone transportation is insufficient [24].

This study systematically evaluates the timeliness and stability of drone medical logistics in China's multi-campus hospital environments for the first time, utilizing 750 real-world flight data points. It innovatively incorporates multi-navigation software comparisons, round-the-clock monitoring, and large-effect-size analysis, providing high-quality evidence for the practical application of drone medical logistics.

## 1.3. Research objectives and questions

This study aims to address the following key questions: (1) How does the timeliness of drone-based medical logistics transportation perform across multiple campuses of urban hospitals? (2) Does the time advantage of drone transportation differ across various time periods when compared to traditional road transportation? (3) Does the stability and reliability of the drone medical logistics system meet clinical demands? By answering these questions, this research seeks to provide scientific evidence and practical guidance for logistics management across multiple campuses in China's public hospitals.

## 2. Materials and methods

### 2.1. Study design and ethical approval

This study constitutes a retrospective systematic evaluation and analysis based on hospital operational data. It does not involve any clinical trials involving human or animal subjects, nor does it collect or utilize any personally identifiable information (including personal details of patients, medical staff, or other participants). All analyzed data pertains exclusively to project operational metrics, such as flight frequencies, transport durations, and cost data. Therefore, in accordance with the exemption provisions outlined in the World Medical Association's Declaration of Helsinki and China's Ethical Review Measures for Biomedical Research Involving Human Subjects, the Ethics Committee of Deyang People's Hospital has reviewed and confirmed that this study is exempt from ethical review (Ethics Review Number: 2024-03-004-K01). All data processing procedures in this study strictly adhere to the hospital's management regulations concerning data security and privacy protection.

## 2.2. Research location and time

The study sites were two campuses of Deyang People's Hospital in Sichuan Province: the Jinghu Campus (main campus) and the Women's and Children's Campus, separated by a straight-line distance of approximately 6.2 kilometers. The study period ran from April 8, 2024, to July 30, 2024, totaling 114 days.

## 2.3. UAV systems and flight plans

### 2.3.1. UAV equipment.
The TR7S intelligent logistics UAV (Fig 1) measures 1110 × 1272 × 450 mm, with a 400 × 300 × 240 mm cargo compartment. Technical specifications include: maximum payload 4.13 kg, cruise speed 60 km/h, operational range 15 km, and service ceiling 2,000 m. During the study period, in order to minimize the differential impact of weather on the results, the intelligent UAV chooses sunny or cloudy days without strong winds, rainfall and other abnormal weather for each cross-institutional logistics and transportation mission, and the researchers declared to the governmental departments in advance and obtained approval. The intelligent logistics UAV has an automated UAV operation and control platform deployed in the cloud, which includes real-time monitoring, remote control, route planning and design, equipment management and maintenance, flight-related data statistics, and other functions required for operation. In order to ensure flight safety, the UAV is designed with a multi-system redundancy backup scheme, including sensor and communication redundancy. Meanwhile, the UAV has an automatic inspection function, which can realize rapid fault identification before takeoff and during flight, fault warning and prohibit flight operation, reducing the operation risk of the UAV.

### 2.3.2. Flight route planning.
The flight route, approved by relevant airspace management authorities, avoids densely populated areas and airport clearance zones. The starting point is the designated landing platform at the Women and Children's Hospital Campus (31°7'23.4" N, 104°23'15.6" E), and terminates at the takeoff/landing platform of the Jinghu

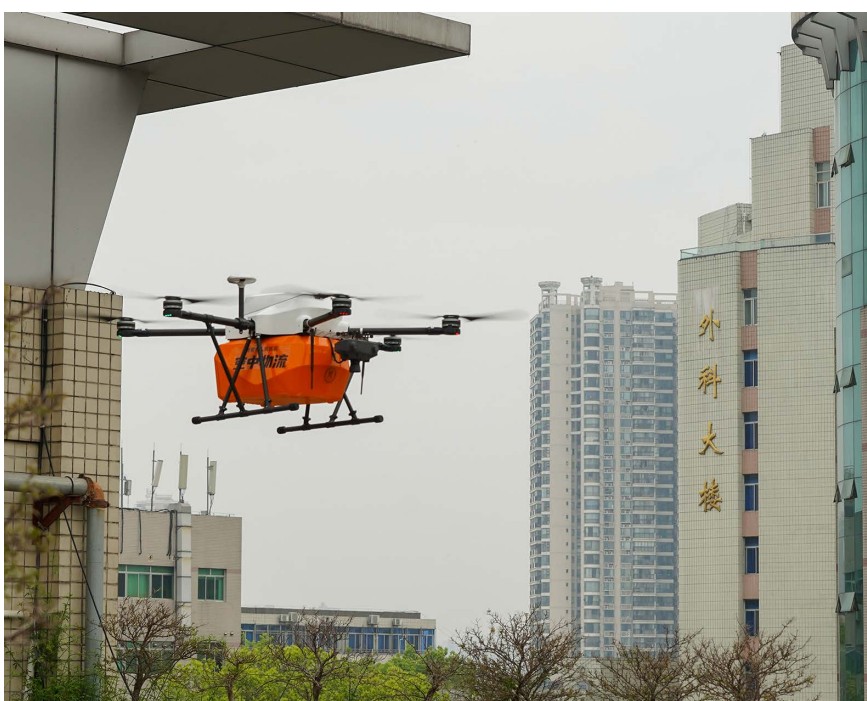

**Fig 1. Intelligent Logistics UAV.**

Campus (31°9'45.2" N, 104°23'8.7" E). The straight-line distance is 5.95 km, with a flight altitude of 120 m and a cruising speed of 16 m/s.

### 2.3.3. Operational procedures.
Drone transport operations strictly adhere to standardized procedures:

(1) Mission Request: Submit transport requests specifying cargo type, priority level, and time constraints.

(2) Equipment Inspection: Conduct pre-flight system checks including battery status (>80% charge required), sensor calibration, and communication signal strength.

(3) Cargo Loading: Utilize specialized transport containers compliant with biological transport standards.

(4) Flight Monitoring: Ground control center conducts real-time monitoring of flight status.

(5) Receipt Confirmation: Upon delivery, confirm receipt via PDA scanning; data automatically uploads to the hospital logistics management system.

## 2.4. Data sources and collection

The primary data sources for this study include drone flight data and ground-based road travel times. The former was directly exported from the project's drone management system, while the latter was obtained by measuring travel times between the two campuses using navigation software.

Acquisition and Compliance Statement for Ground Road Traffic Time Benchmarks. To establish a benchmark for comparing delivery timelines with drone transportation, this study manually queried and recorded predicted road travel times under actual traffic conditions for routes between the same starting and ending campuses during the drone test period. This data was obtained by accessing the publicly available, free real-time route planning and navigation features provided by Baidu Maps, Amap, and Tencent Map through their official mobile applications. These functions constitute standardized, free services offered to the public. We hereby confirm that this data acquisition method fully complies with the terms regarding personal, non-commercial, and reasonable use outlined in the platforms' publicly available Terms of Service. The collected data is exclusively used for comparative analysis within this academic research and has not been utilized for any commercial purposes or bulk scraping.

To assess the efficiency of drone logistics transportation and eliminate the interference of transportation distance differences on timeliness evaluations, this study defines the key indicator of "unit time", which is the total transportation time divided by the transportation distance (min/km), to enable standardized comparisons between different transportation modes (air vs. ground) over the same distance. Specific indicators include:

### 2.4.1. UAV flight data.
Drone flight unit time: Complete data for each flight is extracted from the drone management system, including flight distance (accurate to 0.01 km) and flight time (total time from takeoff to landing, accurate to the second), to calculate the unit time for each flight (min/km).

### 2.4.2. Ground road traffic volume per unit time.
Use mainstream navigation software (Baidu Maps, Amap, Tencent Map) to obtain the shortest ground driving route distance and required time between the same hospital campuses [25–27]. To comprehensively reflect the impact of road conditions at different times on transportation time, ground transportation unit time is measured under the following scenarios:

Midnight baseline period (0:00): Measure ground transportation unit time using Baidu Maps and Amap at midnight (0:00) every day for 30 consecutive days to obtain baseline values under smooth road conditions.

Full-time monitoring: Over 30 consecutive days, ground transportation unit time was measured using Tencent Map at nine time points each day (midnight, 8 AM, 10 AM, 12 PM, 2 PM, 4 PM, 6 PM, 8 PM, and 10 PM) to cover varying traffic flow conditions throughout the day and night, including morning and evening rush hours and nighttime.

Each measurement is repeated three times, with the average value taken as the final result. To ensure data reliability, avoid national statutory holidays, extreme weather conditions (such as heavy rain or dense fog), and days with special traffic restrictions.

## 2.5. Statistical analysis

Data analysis was performed using SPSS 27 statistical software. Quantitative data are presented as mean±standard deviation (Mean±SD) or median±interquartile range (M±IQR). First, a normality test (Shapiro-Wilk test) was conducted on the data. The results showed that the unit time of unmanned aerial vehicles (UAVs) and the unit time of ground traffic at each time point did not follow a normal distribution (P<0.001). Therefore, nonparametric tests were used for between-group comparisons. Specifically, the Mann-Whitney U test was performed to compare the unit time of drones with the unit time of ground traffic at each time point to determine whether there were significant differences between them. To account for multiple comparisons, the Bonferroni method was applied to adjust the significance level ($\alpha$=0.05/9=0.0056) for multiple comparisons, thereby controlling the risk of Type I error. Effect sizes were assessed using Cliff's Delta coefficient. According to Romano et al. [28] standards: $|\delta|$<0.147 indicates a small effect, 0.147–0.33 denotes a moderate effect, 0.33–0.474 signifies a large effect, and $|\delta|$>0.474 represents a very large effect. All statistical tests were two-sided, with P<0.05 (adjusted P<0.0056) indicating a statistically significant difference.

## 3. Results

This study focuses on the evaluation of the effectiveness of drone medical logistics between Deyang City People's Hospital's Maternity and Pediatric Hospital District and Jinghu Hospital District. The average flight distance of UAVs between the two hospital districts is (5.95±0.11) km, which is highly close to the road traffic distance of 6.2 km. From April 8, 2024 to July 30, 2024, a total of 751 missions were conducted between the Women's and Children's Hospital District and the Jinghu Hospital district using drones for logistics transfer. Because there are some differences in the flight distance of each UAV, it is necessary to screen the UAV data to ensure that the flight distance is close to the distance of road transportation (±0.3 km error is allowed), and finally include the valid flight data of 750 flights. In order to eliminate the interference of the difference in transport distance on the timeliness assessment, the index of "unit time (total transport time/transport distance, min/km)" was innovatively proposed to realize the standardized comparison of the transport efficiency between UAVs and roads.

### 3.1. Basic information on drone flight

The drone's flight distance was (5.95±0.03) km, with a flight duration of (9.77±0.87) minutes. The histogram of the distribution of UAV flight unit time is shown in Fig 2, with a median of 1.58 min/km (IQR=0.10) and a mean of 1.64±0.14 min/km. 75% of the flights unit time were concentrated below 1.67 min/km, with a concentrated distribution and small fluctuation (SD=0.14).

### 3.2. Comparison of drone and zero-point road traffic times

The Comparison of the drone flight unit time with road traffic unit time at 0:00 determined by the three common mapping software is shown in Table 1. The unit time measured by the drone was significantly lower than that determined by Baidu Maps software at midnight (0:00) (2.06±0.12 min/km, P<0.001), Amap (2.01±0.12 min/km, P<0.001), Tencent Map (2.03±0.09) min/km (P<0.001). The bar chart comparing drone unit time with the three navigation software units at midnight is shown in Fig 3. The highly consistent measurement results among the three navigation software indicate good reliability of the road traffic data. By comparing the data of the three mainstream maps, the time advantage of the UAV in the baseline time (0:00 a.m.) is presented visually, which lays the foundation for the subsequent full-time analysis.

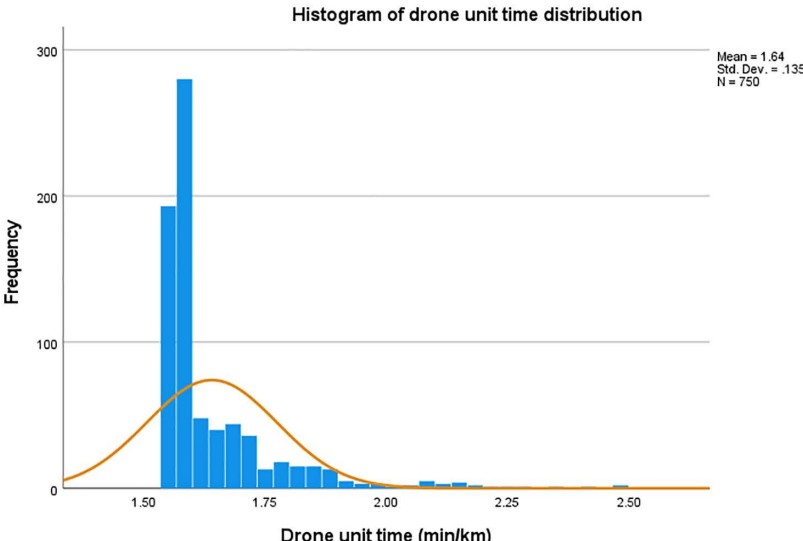

**Fig 2. Histogram of drone unit time distribution.**

**Table 1. Comparison of drone flight distance and unit time with road traffic distance and travel unit time measured by three mainstream navigation map software.**

| logistics | gap (km) | unit time (min/km) | P-value compared with drones | Savings Rate (%) |
|---|---|---|---|---|
| **Drone** | 5.95±0.03 | 1.64±0.14 | – | |
| **Road Traffic (Baidu Maps)** | 6.2 | 2.06±0.12 | < 0.001 | 20.4 |
| **Road traffic (Amap)** | 6.2 | 2.01±0.12 | < 0.001 | 18.4 |
| **Road Traffic (Tencent Map)** | 6.2 | 2.03±0.09 | < 0.001 | 19.2 |

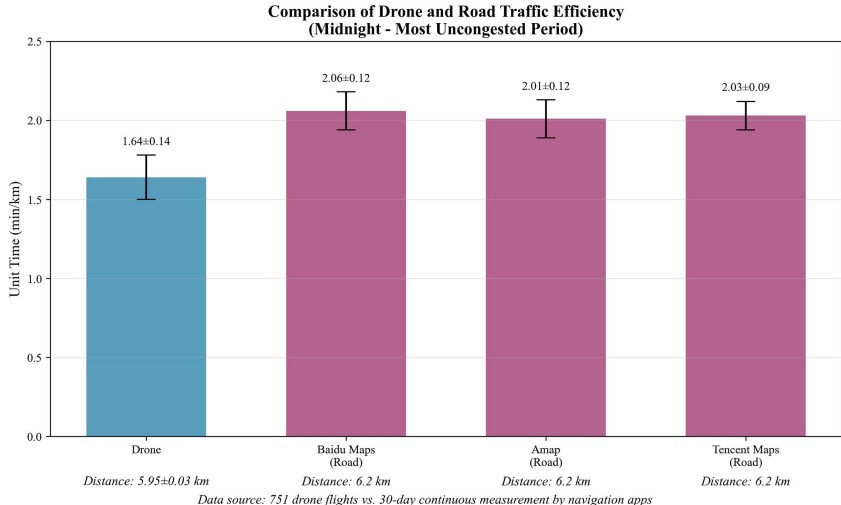

**Fig 3. The bar chart comparing drone unit time with the three navigation software units at midnight.**

### 3.3. Comparison of UAV and road traffic times at different periods

Table 2 shows the road traffic unit time measurement results for nine different time points monitored by Tencent Map over a 30-day period. At nine different time points throughout the day, there is a significant increase in road traffic volume during peak hours (such as 8:00, 10:00, and 18:00). Especially at 10:00, as this time coincides with the peak period for hospital outpatient services, resulting in heavy traffic around hospitals and increased congestion, with road traffic unit time rising to (4.12±0.09) min/km. The next peak periods are the morning and evening commuting hours at 8:00 and 18:00; The smoothest traffic conditions occur at midnight, with a traffic flow rate of (2.03±0.09) min/km. The average flight time for drones is (1.64±0.14) min/km, creating a stark contrast between road traffic and drone stability. Fig 4 shows a line chart comparing the unit time of drones with that of road traffic across nine time points. Therefore, using drones for logistics transportation between the two campuses of the hospital can save up to 15.79 minutes during the 10:00 peak period, representing a savings rate of 60.2%, and a minimum of 2.83 minutes during the midnight smooth traffic period.

**Table 2. Comparison of road traffic unit time between tencent map and drones at nine different time points over 30 consecutive days, along with results of normality tests for the data.**

| times | Road traffic unit time(Mean±SD) (min/km) | Drone unit time (min/km) | Savings Rate (%) | Shapiro-Wilk | *P*-value |
|---|---|---|---|---|---|
| 0:00 | 2.03±0.09 | 1.64±0.14 | 19.2 | 0.720 | <0.001 |
| 8:00 | 3.39±0.12 | 1.64±0.14 | 51.6 | 0.810 | <0.001 |
| 10:00 | 4.12±0.09 | 1.64±0.14 | 60.2 | 0.717 | <0.001 |
| 12:00 | 2.67±0.10 | 1.64±0.14 | 38.6 | 0.742 | <0.001 |
| 14:00 | 2.61±0.12 | 1.64±0.14 | 37.2 | 0.845 | <0.001 |
| 16:00 | 2.65±0.18 | 1.64±0.14 | 38.1 | 0.909 | 0.014 |
| 18:00 | 3.22±0.12 | 1.64±0.14 | 49.1 | 0.811 | <0.001 |
| 20:00 | 2.23±0.14 | 1.64±0.14 | 26.5 | 0.840 | <0.001 |
| 22:00 | 2.09±0.90 | 1.64±0.14 | 21.5 | 0.729 | <0.001 |

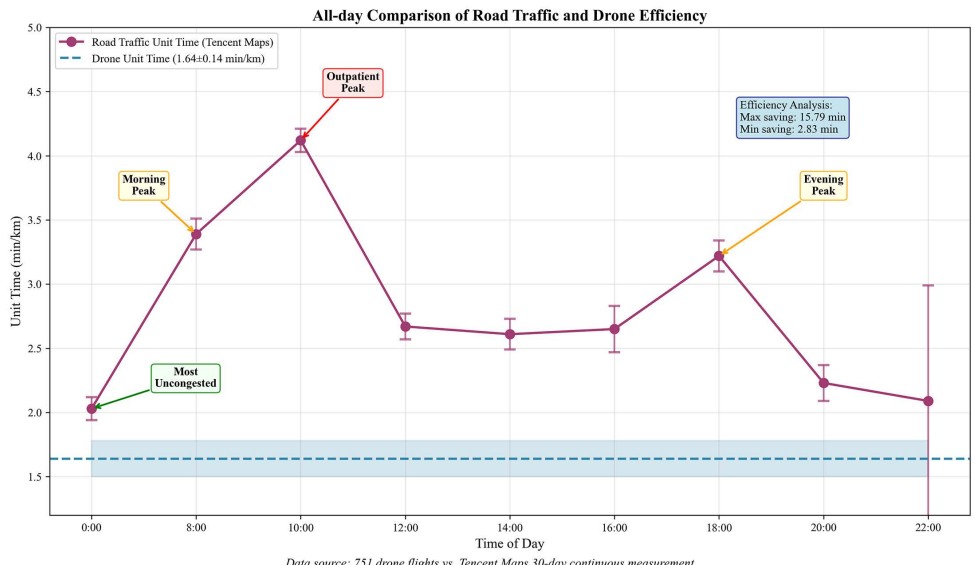

**Fig 4. A line chart comparing the unit time of drones with that of road traffic across nine time points.**

## 3.4. Comparison of timeliness between drones and road transportation

Fig 5 visually compares the unit time distribution of drones and road traffic at different time points using box plots. It can be seen that the box for drone flight unit time is entirely below the box for road traffic unit time at all time points, and at least 75% of drone flight unit time is significantly less than any road traffic unit time measured at any time point, demonstrating the time efficiency advantage of drones across all time periods. Notably, the interquartile range (IQR) of road traffic distribution over time is significantly greater than that of drones, indicating that road traffic exhibits greater temporal variability, while drones demonstrate higher stability. Furthermore, the median difference in drone unit time is significant, and inferential statistical analysis is then used to verify whether this difference is statistically significant.

In this study, since the unmanned aerial vehicle (UAV) had data from 750 flight missions, to avoid interference from outliers, the "median ± interquartile range (Q3-Q1)" of the UAV's flight time per unit was used to replace the "mean ± standard deviation" in the inferential statistical analysis. First, a normality test (Shapiro-Wilk) was conducted to assess the data distribution. The analysis indicated that the UAV flight time per unit time did not follow a normal distribution (Shapiro-Wilk = 0.652, P < 0.001). The unit time of road traffic at nine different time points also did not follow a normal distribution. The results of the normality test are shown in Table 2. Next, the nonparametric test (Mann-Whitney U) was used for statistical analysis. The original hypothesis H0 was that there is no difference in the median of the unit time between drones and road traffic; the alternative hypothesis H1 was that the median of the unit time for drones was significantly smaller.

Merge the drone unit time data with the road traffic unit time data from different time points into the same data file. Use the nonparametric Mann-Whitney U test, repeated nine times, to compare whether there are significant differences between the drone unit time and the road traffic unit time at each time point. Since nonparametric tests involve multiple comparisons, which can lead to Type I statistical errors, the Bonferroni correction was applied to adjust the significance threshold to α = 0.0056, strictly controlling the risk of false positives. After nine comparisons, the Mann-Whitney U test results showed that the "progressive significance (two-tailed) <0.001," with all p-values significantly below the corrected significance level of 0.0056. This strongly rejects the null hypothesis, indicating that there is a significant difference in unit time between drones and road traffic. Additionally, the non-parametric effect size (Cliff's Delta) was used to further

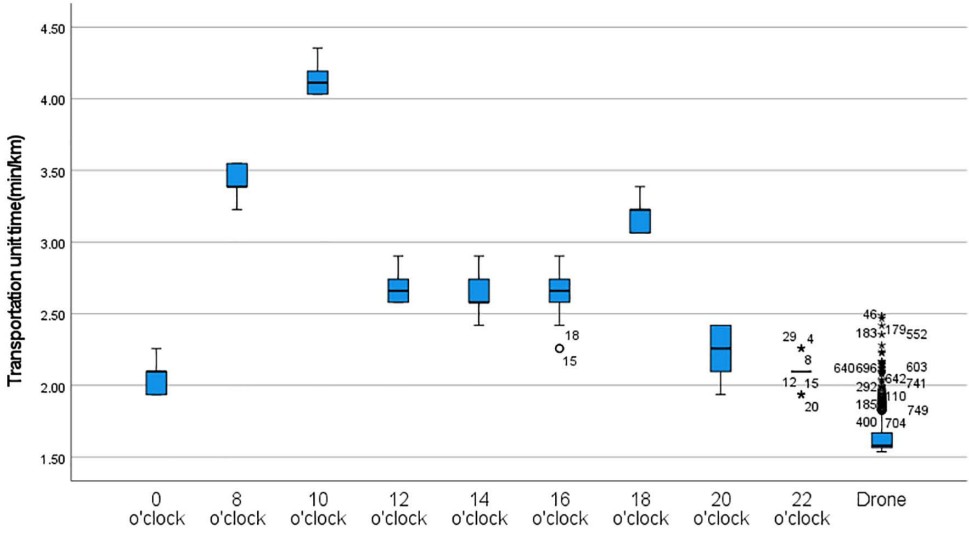

**Fig 5. Box blots of road traffic unit time compared to UAV unit time at different points in time.**

**Table 3. Results of nonparametric tests of UAV unit time versus road traffic unit time at each time point.**

| Times | Drone unit time (min/km) (median±IQR) | Road traffic unit time (min/km) (Mean±SD) | P-value (after calibration) | Cliff's Delta | Results |
|---|---|---|---|---|---|
| **0:00** | 1.58±0.10 | 2.03±0.09 | <0.001 | −0.86 | Great effect, drones are significantly faster |
| **8:00** | 1.58±0.10 | 3.39±0.12 | <0.001 | −0.83 | Great effect, drones are significantly faster |
| **10:00** | 1.58±0.10 | 4.12±0.09 | <0.001 | −0.89 | Great effect, drones are significantly faster |
| **12:00** | 1.58±0.10 | 2.67±0.10 | <0.001 | −0.75 | Big effect, drones are significantly faster |
| **14:00** | 1.58±0.10 | 2.61±0.12 | <0.001 | −0.78 | Big effect, drones are significantly faster |
| **16:00** | 1.58±0.10 | 2.65±0.18 | <0.001 | −0.81 | Great effect, drones are significantly faster |
| **18:00** | 1.58±0.10 | 3.22±0.12 | <0.001 | −0.85 | Great effect, drones are significantly faster |
| **20:00** | 1.58±0.10 | 2.23±0.14 | <0.001 | −0.82 | Great effect, drones are significantly faster |
| **22:00** | 1.58±0.10 | 2.09±0.90 | <0.001 | −0.80 | Great effect, drones are significantly faster |

illustrate the practical significance. However, due to the large sample size of drones, manual calculation of the non-parametric effect size was impractical. This study utilized Python programming tools to calculate Cliff's Delta values, with results shown in Table 3. The Cliff's Delta values for drone unit time compared to road traffic unit time at different time points were all negative, with absolute values exceeding 0.75, indicating that drone transportation unit time is significantly shorter than road transportation unit time, and the effect size is extremely large ($|\delta| > 0.43$ is considered a large effect). This further supports the results of the Mann-Whitney U test, confirming that drone logistics transportation has significant advantages in terms of timeliness and stability, particularly during peak traffic hours.

## 4. Discussion

Through systematic analysis of real-world flight data from 750 missions, this study demonstrates for the first time the dual significant advantages of drone-based medical logistics in terms of timeliness and stability within the multi-campus operational model of urban public hospitals in China. The findings provide high-quality empirical evidence for optimizing medical resource allocation and establishing an intelligent medical logistics system, holding important theoretical and practical significance for enhancing healthcare service efficiency and improving public health emergency response.

### 4.1. Interpretation of key findings

**4.1.1. Advantages in timeliness and stability of UAV medical logistics.** The core finding reveals that the average unit time for cross-campus medical supply transport via UAV was (1.64±0.14) min/km, significantly lower than Baidu Maps (2.06±0.12) min/km, Amap (2.01±0.12) min/km, and Tencent Map (2.03±0.09) min/km for zero-point road traffic unit time (all P<0.001). More importantly, this time advantage exhibited significant variations across different time periods. During the peak traffic period at 10:00, road transportation reached (4.12±0.09) min/km, while drone transportation maintained stable efficiency, achieving a time savings rate as high as 60.2% ($\delta = -0.89$, extremely large effect size). This outcome validates the unique value of drones in complex urban traffic environments, not only providing fundamental transportation functions but also ensuring timely delivery of medical supplies during critical periods.

The high stability of drone transportation is equally noteworthy. In this study, the standard deviation of drone time per unit distance was only 0.14 min/km, far smaller than the variability observed for ground transportation across all time periods, demonstrating exceptional consistency. This consistency enables drone logistics systems to deliver predictable, reliable transportation services, effectively mitigating the risk of medical delays caused by unpredictable ground traffic conditions. This characteristic is crucial for transporting time-sensitive medical supplies, such as emergency test specimens, intraoperative frozen sections, and thrombolytic drugs, directly impacting the timeliness of clinical decisions and patient outcomes.

**4.1.2. Mechanism analysis of time advantage.** The time advantage of drone transportation primarily stems from the synergistic effects of three factors. First, the path optimization effect, where the drone flight distance (5.95 km) is 4.0% shorter than the road transport distance (6.2 km), enabling near-straight-line point-to-point delivery. Second, the constant speed effect, with the drone maintaining a stable cruising speed of 16 m/s (57.6 km/h), unaffected by traffic signals, congested areas, or pedestrian interference. Third, the process simplification effect: the drone's vertical takeoff and landing capability eliminates transfer waiting times inherent in traditional ground transportation, enabling an "on-demand" instant response model. These three factors collectively form the core competitiveness of drone medical logistics, demonstrating superior time efficiency over conventional transportation methods in urban environments.

## 4.2. Comparison with existing literature

The findings of this study align with relevant international research while also introducing significant innovations. Similar to Homier et al.'s [16] findings in rural African settings, we observed that drone transport was significantly faster than ground transport. However, the time savings observed in this study (19.2%–60.2%) were substantially higher than the 15%–35% reported by Homier et al. This discrepancy is primarily attributed to the marked differences in traffic congestion between urban and rural environments. Studies within the UK's NHS system indicate that introducing drones can reduce sample transport times by approximately 70% (weather permitting) [29,30]. This is comparable to the peak-period efficiency gains (60.2%) observed in this study, further validating the universal value of drones in medical logistics.

Compared to existing studies [31,32], this research offers the following advantages: (1) larger sample size (750 flights), enhancing statistical power; (2) longer observation period (114 days), better reflecting long-term operational outcomes; (3) cross-validation using multiple navigation software (Baidu Maps, Amap, Tencent Map), improving data reliability; (4) Incorporation of nonparametric effect size analysis (Cliff's Delta) for a more comprehensive assessment of clinical significance. These methodological strengths enhance the scientific rigor and persuasiveness of our findings.

The core innovation of this study manifests in three dimensions. First, it represents the inaugural large-scale real-world study conducted within the specific context of multi-site urban hospitals in China, filling a gap in the field. Previous studies predominantly focused on remote areas or simulated environments [16,33], whereas this research centers on densely populated, traffic-complex urban core zones, yielding findings with greater clinical translation value. Second, it systematically analyzed transportation efficiency differences across time periods, revealing the unique value of drones during peak traffic hours and providing data support for dynamic allocation of hospital logistics resources. Third, it detailed the integration process between UAVs and hospital information systems, covering critical steps such as mission requests, flight monitoring, and data traceability, offering an actionable reference for practical implementation. These innovations directly address the academic call for "more real-world studies to validate the clinical value of UAV medical applications."

## 4.3. Clinical and public health significance

**4.3.1. Enhancing emergency treatment efficiency.** Inter-hospital rapid logistics are critical for improving the efficiency of emergency patient care. Traditional manual specimen transport often involves lengthy waiting times and lacks real-time tracking mechanisms, frequently leading to delayed diagnostic reports. Given the significant time advantages identified in this study, drone logistics systems hold promise as a key technological enabler for boosting emergency treatment efficiency. Specifically, its clinical value manifests in: (1) Reducing Specimen Turnaround Time (TAT). Studies [34–36] demonstrate that shorter TAT significantly improves patient outcomes. As a key performance indicator for healthcare system efficiency and quality, optimizing TAT effectively reduces mortality, minimizes complications, and accelerates treatment initiation. (2) Optimizing intraoperative frozen section pathology workflows. Rapid specimen transport reduces surgical waiting times and lowers anesthesia risks. (3) Enabling "minute-level" delivery of emergency medications, securing the "golden hour" for treating time-dependent conditions like stroke and myocardial infarction.

**4.3.2. Value in enhancing public health emergency response systems.** During public health emergencies (e.g., major epidemics, natural disasters, accidents), the urgent allocation of medical supplies and rapid sample testing are critical for controlling the situation. During the 2020 COVID-19 pandemic, traditional logistics systems exposed transportation bottlenecks, while drones—unaffected by ground traffic disruptions—emerged as an ideal emergency response solution [37]. The standardized operating procedures and quality control system established in this study provide a replicable and scalable model for medical supply allocation during emergencies like epidemics. For instance, drones can rapidly centralize sample delivery for testing during large-scale nucleic acid testing campaigns. At public emergency sites, they can swiftly deliver emergency medications and small equipment, enhancing on-site treatment capabilities. Thus, the UAV medical logistics system serves not only as an optimization tool for hospital operations but also as a vital component in building a resilient public health emergency response framework.

**4.3.3. Health economics evaluation.** From a health economics perspective, while UAV systems require higher initial investment, they demonstrate significant advantages in long-term operational costs. Based on this study's data, the unit cost of drone transportation is approximately 61 yuan per flight, while traditional ground vehicle transportation (including labor costs, vehicle depreciation, and fuel expenses) costs about 80 yuan per trip. Calculating at 10 flights per day and 3,650 flights per year, adopting drone logistics could save approximately 69,350 yuan in annual operating costs. Notably, this cost advantage will further expand with technological advancements (improved battery endurance, increased automation) and economies of scale (higher transport frequency). Additionally, while indirect benefits from drone transport, such as reduced emergency room dwell time, lower reoperation rates, and improved bed turnover, are not included in direct cost calculations, their contribution to enhancing overall hospital operational efficiency hold significant economic value.

## 4.4. Limitations and future research directions

**4.4.1. Limitations of this study.** Although this study yielded valuable findings, several limitations warrant clarification: First, the single-center study design limits the generalizability of results. Conducted exclusively at Deyang People's Hospital in Sichuan Province, its geographical environment, traffic conditions, and hospital layout may differ from other regions. Future multi-center, multi-regional studies are needed for validation. Second, road travel time data was derived from navigation software simulations rather than actual driving records. Although cross-validation using three independent software programs enhanced reliability, discrepancies with real-world transportation times may still exist. Third, weather conditions imposed limitations: flight data during severe weather events such as strong winds or heavy rain was excluded, preventing a comprehensive assessment of UAV performance under extreme conditions. Fourth, the integration depth of information systems was insufficient. This study primarily focused on transportation time metrics without delving into the issue of deep data interaction between the drone logistics system and existing hospital information systems (e.g., electronic medical record systems, laboratory information systems, hospital resource planning systems). Finally, long-term effect evaluation was lacking. The observation period of 114 days was too short to assess long-term indicators such as the stability of the drone system, equipment depreciation, and maintenance costs.

**4.4.2. Future research directions.** Based on these limitations and current research frontiers, future studies should prioritize the following directions.

Life-cycle cost-benefit analysis. Conduct long-term tracking studies and implement more real-world, large-scale demonstration projects to collect key performance and cost data. Conduct a comprehensive evaluation of the full-cycle costs of drone logistics systems, including initial investment, equipment depreciation, energy consumption costs, maintenance expenses, and personnel training. Perform a thorough cost-effectiveness and cost-utility analysis comparing these systems with traditional transportation methods.

Optimization of Specific Medical Supply Transportation Solutions. Develop specialized transport modules and temperature-controlled systems tailored to the characteristics of different medical supplies (e.g., blood products, vaccines,

biological samples, emergency medications) to ensure quality and safety, particularly focusing on stability research for temperature-sensitive pharmaceuticals and biological agents.

Collaborative Design of UAV Logistics Networks. Explore multi-site collaborative transport models, investigate path optimization algorithms, task scheduling strategies, and emergency response mechanisms for UAV logistics networks, achieving a transition from "point-to-point" transport to "networked" services.

Regulatory Framework and Standards System Research [7]. Aligned with China's airspace management characteristics, explore UAV operational regulations and standards suitable for medical scenarios, including flight safety specifications, privacy protection measures, data security standards, and liability determination mechanisms, providing policy support for the industry's healthy development.

Emerging Technology Integration Applications. Investigate the application of 5G communication technology in UAV remote control and real-time monitoring; explore the optimization role of artificial intelligence algorithms in dynamic flight path planning and obstacle avoidance; evaluate the value of blockchain technology in medical supply traceability and supply chain management [38–40].

Environmental Adaptability Research. Systematically evaluate the impact of varying weather conditions (temperature, humidity, wind speed, precipitation) on drone flight performance and delivery times. Establish environmental risk assessment models and contingency plans.

### 4.5. Practical application recommendations

Based on the findings of this study and practical operational experience, we propose the following practical application recommendations for reference by medical institutions and relevant authorities.

Selection of Applicable Scenarios. For hospitals operating a multi-campus model, it is recommended to prioritize deploying drone logistics systems between campuses located 5–15 km apart. This distance range maximizes the time-sensitive advantages of drones while avoiding endurance issues caused by excessively long flight distances.

Establishment of Standardized Management Systems. Develop a comprehensive UAV logistics management framework encompassing airspace application procedures, routine equipment maintenance protocols, flight safety checklists, contingency plans, and quality control standards to ensure secure and stable system operation.

Focus on Key Application Areas. Initially concentrate on transporting time-sensitive medical supplies such as emergency lab specimens, intraoperative frozen sections, critical medications, and blood products to rapidly achieve clinical value and economic benefits.

Technological Collaborative Innovation. Partner with navigation software companies to develop specialized traffic prediction models for medical logistics, optimizing drone dispatch strategies using hospital operational data. Strengthen technical collaboration with drone manufacturers to customize equipment and functional modules tailored for medical scenarios.

Policy and Regulatory Engagement. Actively communicate with local airspace management authorities and health administration departments to participate in establishing regional standards and industry norms for drone-based medical logistics, fostering a favorable policy environment for technology adoption [41].

Multidisciplinary Team Development. Establish a multidisciplinary team comprising clinical physicians, laboratory technicians, logistics management experts, aviation technicians, and information engineers to jointly advance the clinical translation and continuous optimization of drone-based medical logistics.

### 5. Conclusion

This study systematically demonstrated for the first time, through large-scale real-world data, the significant value of drones in medical logistics transportation across multiple campuses of urban hospitals. Drones not only offered faster delivery speeds than traditional road transportation, particularly during peak traffic hours, but also exhibited greater

stability, which was crucial for ensuring timely and reliable delivery of medical supplies. The findings provided compelling evidence for optimizing multi-campus logistics management in China's public hospitals, supporting the widespread adoption of drones as an innovative technological solution in healthcare.

With continuous advancements in drone technology and the gradual refinement of regulatory frameworks, we believe drone-based medical logistics will become an integral component of smart hospital development. This technology holds great promise for enhancing healthcare service efficiency, improving patient outcomes, and strengthening public health emergency response systems. Moving forward, collaborative efforts among industry, academia, and government agencies are essential to foster the healthy development and standardized application of this innovative technology.

## Supporting information

**S1 Table. Road traffic travel time measured by Tencent Map at nine time points and UAV-related flight data.**
(XLSX)

**S2 Table. Road traffic travel time measured by Tencent Map at nine time points.**
(XLSX)

**S3 Table. Road traffic travel time measured by three navigation software programs at 0 o'clock.**
(XLSX)

**S4 Table. UAV-related flight data.**
(XLSX)

**S5 Code. Data Visualization Code for Figures 3 and 4.**
(DOCX)

## Author contributions

**Conceptualization:** Xingbo Long.

**Data curation:** Xingbo Long, Min Huang.

**Formal analysis:** Min Huang, Xiaojiang Yu.

**Funding acquisition:** Xingbo Long, Xiankai Xie.

**Investigation:** Xingbo Long.

**Project administration:** Xiankai Xie, Yanxin Wang, Wei Zhou, Bo Hu.

**Resources:** Xiankai Xie, Wei Zhou.

**Supervision:** Xiankai Xie.

**Visualization:** Xingbo Long, Min Huang.

**Writing – original draft:** Xingbo Long, Min Huang.

**Writing – review & editing:** Min Huang, Xiankai Xie, Yanxin Wang.

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
