## [Decision Letter · Decision Letter 0]

4 Nov 2025

PONE-D-25-54130An empirical study of drone medical logistics transportation in a multi-campus model of Chinese public hospitals: real-world data-driven validation of timeliness and application effectsPLOS ONE

Dear Dr. Long,

Thank you for submitting your manuscript to PLOS ONE. After careful consideration, we feel that it has merit but does not fully meet PLOS ONE’s publication criteria as it currently stands. Therefore, we invite you to submit a revised version of the manuscript that addresses the points raised during the review process.

We look forward to receiving your revised manuscript.

Kind regards,

Zeashan Hameed Khan, Ph.D.

Academic Editor

PLOS ONE

Journal Requirements:

2. In your Methods section, please include additional information about your dataset and ensure that you have included a statement specifying whether the collection and analysis method complied with the terms and conditions for the source of the data.

3. Please note that PLOS One has specific guidelines on code sharing for submissions in which author-generated code underpins the findings in the manuscript. In these cases, we expect all author-generated code to be made available without restrictions upon publication of the work. Please review our guidelines at https://journals.plos.org/plosone/s/materials-and-software-sharing#loc-sharing-code and ensure that your code is shared in a way that follows best practice and facilitates reproducibility and reuse.

4. In your Methods section, please provide additional information regarding the permits you obtained for the work. Please ensure you have included the full name of the authority that approved the field site access and, if no permits were required, a brief statement explaining why.

“This work was supported by National Institute of Hospital Administration [grant number 2025RWE023]; Sichuan Province Health Information Center [grant number 2021ZXKY06012]; Sichuan Province Health Information Society [grant number 2022051]; and Sichuan Provincial Hospital Association [grant number SCZB013].”

“This work was supported by National Institute of Hospital Administration [grant number 2025RWE023]; Sichuan Province Health Information Center [grant number 2021ZXKY06012]; Sichuan Province Health Information Society [grant number 2022051]; and Sichuan Provincial Hospital Association [grant number SCZB013]”

“This work was supported by National Institute of Hospital Administration [grant number 2025RWE023]; Sichuan Province Health Information Center [grant number 2021ZXKY06012]; Sichuan Province Health Information Society [grant number 2022051]; and Sichuan Provincial Hospital Association [grant number SCZB013].”

Additional Editor Comments:

This paper describes an empirical study of drone medical logistics transportation in a multi-campus model of Chinese public hospitals by evaluating the real-world data-driven validation of timeliness and application effects. The paper requires significant improvements before it can be further considered for publication in PLOS One.

Reviewer's Responses to Questions

**Comments to the Author**

1. Is the manuscript technically sound, and do the data support the conclusions?

Reviewer #1: Partly

Reviewer #2: Yes

2. Has the statistical analysis been performed appropriately and rigorously? 

Reviewer #1: No

Reviewer #2: Yes

3. Have the authors made all data underlying the findings in their manuscript fully available?

Reviewer #1: Yes

Reviewer #2: Yes

4. Is the manuscript presented in an intelligible fashion and written in standard English?

Reviewer #1: Yes

Reviewer #2: Yes

5. Review Comments to the Author

Reviewer #1: 1. The abstract includes excessive numerical data and statistical details, which make it difficult to follow and obscure the main findings. The authors should summarize key performance outcomes concisely and focus on the broader implications of UAV-based logistics rather than listing every numerical result.

2. Although the study claims to “pioneer” UAV feasibility evaluation for medical logistics, it lacks a clear explanation of what distinguishes this work from existing studies in medical UAV transport. The authors should also acknowledge methodological or operational limitations—such as regulatory, weather, or payload constraints—to make the abstract more balanced and realistic.

3. The introduction should clearly conclude with a distinct paragraph that highlights the novel contributions of your work.

4. The literature review should benefit from more explorations of previous studies.

5. The discussion section needs to be expanded to more thoroughly analyze the results.

6. The first paragraph of the conclusion should succinctly summarize the contributions of the study in past tense.

7. The second paragraph of the conclusion should provide clear and actionable future recommendations.

8. Some equations are not properly cited.

9. Visualizations from Table 3 would add good value to the manuscript.

Reviewer #2: Paper Review and Comments

1. Title and Abstract

• Title: An empirical study of drone medical logistics transportation in a multi-campus model of Chinese public hospitals: real-world data-driven validation of timeliness and application effects

The introduction section should provide a clearer explanation of the aim and motivation of the study, as the current version is brief and lacks sufficient context. Additional references and detailed explanations should be incorporated to strengthen the introduction. The research objectives are not explicitly stated and should be clearly outlined to guide readers regarding the purpose and focus of the paper. The related work section could be enhanced by including more relevant and recent studies to provide a stronger foundation for the research.

The conclusion section could be improved by providing a more detailed summary of the findings and highlighting the key contributions of the study. While the paper is generally readable, minor grammatical issues should be revised to improve clarity and overall presentation. The reference list is limited and should be extended with additional relevant and recent works, and the formatting of references should be made consistent and professional.

Figures and tables should be discussed in greater detail within the text to improve interpretation and context. All figures should be properly referred to in the text and briefly explained. Tables should have their captions positioned above the table, and the content should be explained in more detail for clarity. Each equation or algorithm should be fully explained, including a description of all parameters used.

The introduction could benefit from incorporating the following references to enhance depth and recency:

A. J. Moshayedi, Y. Xie, M. Sharifdoust, and A. S. Khan, "Evaluating OMNI robot navigation with SLAM in CoppeliaSim: Hemangiomas and nonhomogeneous paths," J. Robot. Res. (JRR), vol. 1, no. 1, pp. 7–14;

10.1109/ACCESS.2024.3355278;

doi:10.5829/ije.2025.38.07a.07.

Additionally, the authors are encouraged to address the following questions to clarify and strengthen the paper:

• What are the main objectives of the paper?

• How did environmental factors such as weather conditions, wind speed, or visibility affect UAV performance and flight time during the study period?

• What are the main technical or regulatory challenges that currently limit large-scale deployment of UAV logistics in public hospitals across China?

• Could you elaborate on how the proposed UAV logistics framework could be integrated with existing hospital information systems to ensure seamless coordination and real-time tracking of medical materials?

Overall, while the study presents valuable insights, the paper format and presentation should be revised according to the conference template. Major revisions are recommended before acceptance.

6. PLOS authors have the option to publish the peer review history of their article (what does this mean?). If published, this will include your full peer review and any attached files.

Reviewer #1: No

Reviewer #2: No

---

## [Author Response · Author response to Decision Letter 1]

23 Jan 2026

Point-by-point response to editors and reviewers’ comments

Dear Editors and Reviewers

Thank you for your valuable suggestions and for the reviewers' comments concerning our manuscript entitled“An empirical study of drone medical logistics transportation in a multi-campus model of Chinese public hospitals: real-world data-driven validation of timeliness and application effects”(ID:PONE-D-25-54130). These comments are all valuable and very helpful for revising and improving our manuscript, as well as the important guiding significance to our researches. We have studied comments carefully and have made correction which we hope meet with approval. Revised portion are marked in red in the manuscript. The main corrections in the manuscript and the responds to editors and reviewers’ comments are as following:

Regarding the issues raised on January 14, 2026, point-by-point response to editors:

1 & 2. Regarding the update to the author list and the request for an Authorship Change Form

Response: Thank you for your instructions. We have completed the two required actions regarding authorship for our manuscript (ID: PONE-D-25-54130):

Authorship Change Request Form: The completed forms have been uploaded to the submission system (file type ‘Other’).

Written Confirmation from All Authors: As requested, all co-author have sent their written consent statement directly to your editorial office(plosone@plos.org), confirming their agreement to the restored author list. The compilation of author consent emails was uploaded to file inventory.

For your reference, the corresponding email addresses for all authors are:

Xingbo Long (Corresponding Author): [723766496@qq.com]

Min Huang: [993557884@qq.com]

Xiankai Xie: [kellandx@qq.com(75502266@qq.com)]

Yanxin Wang: [1078439109@qq.com]

Xiaojiang Yu: [822797265@qq.com]

Wei Zhou: [10205242@qq.com]

Bo Hu: [975171797@qq.com]

Thank you again for your time and consideration. We believe our manuscript now fully complies with the journal's policies and are grateful for the opportunity to clarify these points.

Regarding the issues raised on November 5, 2025, point-by-point response to editors:

1.Please ensure that your manuscript meets PLOS ONE's style requirements, including those for file naming. The PLOS ONE style templates can be found at https://journals.plos.org/plosone/s/file?id=wjVg/PLOSOne_formatting_sample_main_body.pdf and https://journals.plos.org/plosone/s/file?id=ba62/PLOSOne_formatting_sample_title_authors_affiliations.pdf

Response: We have thoroughly reformatted the entire manuscript in strict accordance with the formatting template provided by PLOS ONE. Specifically, we utilized the paper template from the link you provided and adjusted the font, line spacing, section headings, figure and table captions, and reference formatting. The revised main manuscript file has been named as requested: Revised Manuscript with Track Changes.docx.

2. In your Methods section, please include additional information about your dataset and ensure that you have included a statement specifying whether the collection and analysis method complied with the terms and conditions for the source of the data.

Response: In the “Data Sources and Collection” subsection of the “Materials and Methods” section, we have supplemented a detailed description: “The primary data for this study comprised drone flight data and ground road travel times. The former was directly exported from the project's drone management system, while the latter was obtained by measuring travel times between the two campuses using navigation software. Acquisition and Compliance Statement for Ground Road Traffic Time Benchmarks. To establish a benchmark for comparing delivery timelines with drone transportation, this study manually queried and recorded predicted road travel times under actual traffic conditions for routes between the same starting and ending campuses during the drone test period. This data was obtained by accessing the publicly available, free real-time route planning and navigation features provided by Baidu Maps, Amap, and Tencent Maps through their official mobile applications. These functions constitute standardized, free services offered to the public. We hereby confirm that this data acquisition method fully complies with the terms regarding personal, non-commercial, and reasonable use outlined in the platforms' publicly available Terms of Service. The collected data is exclusively used for comparative analysis within this academic research and has not been utilized for any commercial purposes or bulk scraping.” Compliance Statement: We confirm that the data acquisition and analysis methods employed in this study fully comply with relevant regulations.

3. Please note that PLOS One has specific guidelines on code sharing for submissions in which author-generated code underpins the findings in the manuscript. In these cases, we expect all author-generated code to be made available without restrictions upon publication of the work. Please review our guidelines at https://journals.plos.org/plosone/s/materials-and-software-sharing#loc-sharing-code and ensure that your code is shared in a way that follows best practice and facilitates reproducibility and reuse.

Response: This study employed custom-written code and utilized Python for data visualization. We have adhered to PLOS ONE's code-sharing mechanism to ensure unrestricted access and reuse. Additionally, we have included a “Data availability” statement and “Supporting information” at the end of the manuscript.

4. In your Methods section, please provide additional information regarding the permits you obtained for the work. Please ensure you have included the full name of the authority that approved the field site access and, if no permits were required, a brief statement explaining why.

Response: We have supplemented the following information in the “Research Design and Ethical Approval” subsection of the “Materials and Methods” section: “This study constitutes a retrospective systematic evaluation and analysis based on hospital operational data. It does not involve any clinical trials involving human or animal subjects, nor does it collect or utilize any personally identifiable information (including personal details of patients, medical staff, or other participants). All analyzed data pertains exclusively to project operational metrics, such as flight frequencies, transport durations, and cost data. Therefore, in accordance with the exemption provisions outlined in the World Medical Association's Declaration of Helsinki and China's Ethical Review Measures for Biomedical Research Involving Human Subjects, the Ethics Committee of Deyang People's Hospital has reviewed and confirmed that this study is exempt from ethical review (Ethics Review Number: 2024-03-004-K01). All data processing procedures in this study strictly adhere to the hospital's management regulations concerning data security and privacy protection.” Therefore, in accordance with relevant regulations, this study did not seek additional ethical committee approval.

“This work was supported by National Institute of Hospital Administration [grant number 2025RWE023]; Sichuan Province Health Information Center [grant number 2021ZXKY06012]; Sichuan Province Health Information Society [grant number 2022051]; and Sichuan Provincial Hospital Association [grant number SCZB013].”

Response: We acknowledge that the specific roles of the funding sources in this study are as follows: National Institute of Hospital Administration, Sichuan Province Health Information Center, Sichuan Province Health Information Society and Sichuan Provincial Hospital Association provided financial support for this study. Beyond this, the funding sources did not participate in the study design, data collection and analysis, publication decisions, or manuscript writing. Please assist in updating the funding declaration in the online submission form to: “This work was supported by National Institute of Hospital Administration [grant number 2025RWE023]; Sichuan Province Health Information Center [grant number 2021ZXKY06012]; Sichuan Province Health Information Society [grant number 2022051]; and Sichuan Provincial Hospital Association [grant number SCZB013]. The funders had no role in study design, data collection and analysis, decision to publish, or preparation of the manuscript.”

6. Thank you for stating the following in the Acknowledgments Section of your manuscript:“This work was supported by National Institute of Hospital Administration [grant number 2025RWE023]; Sichuan Province Health Information Center [grant number 2021ZXKY06012]; Sichuan Province Health Information Society [grant number 2022051]; and Sichuan Provincial Hospital Association [grant number SCZB013]”

“This work was supported by National Institute of Hospital Administration [grant number 2025RWE023]; Sichuan Province Health Information Center [grant number 2021ZXKY06012]; Sichuan Province Health Information Society [grant number 2022051]; and Sichuan Provincial Hospital Association [grant number SCZB013].”

Response: We have fully removed all references to the previously mentioned funding source and grant number from the manuscript as requested. We have included our amended statements within our cover letter. We hereby formally request that the “Funding Statement” in our online submission system be updated to: “This work was supported by National Institute of Hospital Administration [grant number 2025RWE023]; Sichuan Province Health Information Center [grant number 2021ZXKY06012]; Sichuan Province Health Information Society [grant number 2022051]; and Sichuan Provincial Hospital Association [grant number SCZB013].”

Response: We have carefully reviewed the specific literature cited in the reviewer's suggestions. After evaluation, two of the references pertain to robotics: one concerns the design and development of food delivery robots, while the other addresses robotic applications in corn cultivation. These topics differ from the medical logistics drone applications investigated in our study and thus were not included. We appreciate the reviewer's valuable suggestions.

Point-by-point response to reviewers:

Reviewer #1:

1. The abstract includes excessive numerical data and statistical details, which make it difficult to follow and obscure the main findings. The authors should summarize key performance outcomes concisely and focus on the broader implications of UAV-based logistics rather than listing every numerical result.

Response: We fully agree with your perspective. In the revised abstract, we have removed secondary numerical results. Simultaneously, we have strengthened the presentation of the study's primary conclusions and their broader implications for the field of medical logistics using drones, enhancing the abstract's readability and conciseness. Specific modifications can be found on page 1 of the manuscript.

2. Although the study claims to “pioneer” UAV feasibility evaluation for medical logistics, it lacks a clear explanation of what distinguishes this work from existing studies in medical UAV transport. The authors should also acknowledge methodological or operational limitations—such as regulatory, weather, or payload constraints—to make the abstract more balanced and realistic.

Response:

Novelty: In the “1. Introduction” section, we added subsection “1.2.2 Limitations of existing research and innovations of this study,” highlighting how this research differs from existing literature: This study systematically evaluates the timeliness and stability of drone medical logistics in a multi-hospital urban setting in China for the first time, utilizing real-world flight data from 750 missions. It innovatively incorporates comparisons of multiple navigation software, round-the-clock monitoring, and large-effect-size analysis, providing high-quality evidence for the practical application of drone medical logistics. Specific modifications are detailed on pages 5 to 6 of the manuscript.

Limitations: Although this study yielded valuable findings, certain limitations exist. In the “4. Discussion” section, we have added subsection “4.4.1 Limitations of this study” to candidly discuss the boundaries of this research, including but not limited to: the single-center study design limiting the generalizability of results, and the insufficient consideration of weather variation effects. Specific revisions can be found on pages 26 to 27 of the manuscript.

3. The introduction should clearly conclude with a distinct paragraph that highlights the novel contributions of your work.

Response: Indeed, the introduction should include a distinct paragraph summarizing the novel contributions of this research. We have added a separate section titled “1.3 Research Objectives and Questions” at the end of the introduction, clearly outlining the core issues addressed by this study and the anticipated primary academic or practical contributions. This provides readers with a clear research roadmap.

4. The literature review should benefit from more explorations of previous studies.

Response: In the “1. Introduction” section, we have expanded the “Literature Review” or ‘Background’ section. Not only have we increased the number of recent key references, but more importantly, we have strengthened the critical analysis of the existing research landscape. This allows for a more natural introduction to the necessity of this study. Please see the revision in “1.2 Literature Review” on page 4.

5. The discussion section needs to be expanded to more thoroughly analyze the results.

Response: We have substantially rewritten and expanded the Discussion section. The revised discussion:

- No longer merely reiterates results but delves into the underlying causes and mechanisms behind key findings.

- Provides detailed comparisons between our results and those from similar or dissimilar studies reported in the literature, explaining both similarities and differences.

- Explores specific implications of the findings for policymakers, hospital administrators, and technology developers.

Revisions can be found on pages 21–30.

6. The first paragraph of the conclusion should succinctly summarize the contributions of the study in past tense.

Response: We have rewritten the conclusion section. The first paragraph of the conclusion now summarizes the main work completed and core findings of this study in the past tense, clearly and concisely, directly addressing the research objectives. Please see page 30 for the revisions.

7. The second paragraph of the conclusion should provide clear and actionable future recommendations.

Response: In the Discussion section, we have transformed the second paragraph of the Conclusions into a forward-looking “Research Directions” and “Practical Application Recommendations” section. Here, we pr

---

## [Decision Letter · Decision Letter 1]

9 Feb 2026

PONE-D-25-54130R1An empirical study of drone medical logistics transportation in a multi-campus model of Chinese public hospitals: real-world data-driven validation of timeliness and application effectsPLOS One

Dear Dr. Long,

Thank you for submitting your manuscript to PLOS ONE. After careful consideration, we feel that it has merit but does not fully meet PLOS ONE’s publication criteria as it currently stands. Therefore, we invite you to submit a revised version of the manuscript that addresses the points raised during the review process.

We look forward to receiving your revised manuscript.

Kind regards,

Zeashan Hameed Khan, Ph.D.

Academic Editor

PLOS One

Journal Requirements:

Additional Editor Comments:

Thank you for submitting the revised manuscript. Although the major concerns are addressed, some minor corrections are still required.

Reviewers' comments:

Reviewer's Responses to Questions

**Comments to the Author**

1. If the authors have adequately addressed your comments raised in a previous round of review and you feel that this manuscript is now acceptable for publication, you may indicate that here to bypass the “Comments to the Author” section, enter your conflict of interest statement in the “Confidential to Editor” section, and submit your "Accept" recommendation.

Reviewer #1: All comments have been addressed

Reviewer #2: All comments have been addressed

2. Is the manuscript technically sound, and do the data support the conclusions?

Reviewer #1: (No Response)

Reviewer #2: Yes

3. Has the statistical analysis been performed appropriately and rigorously? 

Reviewer #1: (No Response)

Reviewer #2: Yes

4. Have the authors made all data underlying the findings in their manuscript fully available?

Reviewer #1: (No Response)

Reviewer #2: Yes

5. Is the manuscript presented in an intelligible fashion and written in standard English?

Reviewer #1: (No Response)

Reviewer #2: Yes

6. Review Comments to the Author

Reviewer #1: The revisions are accepted, please update the manuscript as it has some typos and needs proofreading.

Reviewer #2: the authors respond to all my concern and can be published ,the authors respond to all my concern and can be published

7. PLOS authors have the option to publish the peer review history of their article (what does this mean?). If published, this will include your full peer review and any attached files.

Reviewer #1: No

Reviewer #2: No

---

## [Author Response · Author response to Decision Letter 2]

15 Feb 2026

Point-by-point response to editors and reviewers’ comments

Dear Editors and Reviewers,

Thank you very much for your positive feedback and for the time and effort you have dedicated to reviewing our manuscript entitled “An empirical study of drone medical logistics transportation in a multi-campus model of Chinese public hospitals: real-world data-driven validation of timeliness and application effects” (ID:PONE-D-25-54130).

We are delighted that both reviewers have indicated that all previous comments have been addressed and that the manuscript is now considered technically sound. Below, we provide a point-by-point response to the remaining minor issues raised in your latest communication.

Response to Journal Requirements:

1. Regarding the recommendation to cite specific previously published works

We have carefully reviewed all publications suggested by the reviewers during the previous rounds of review. Relevant works have not already been incorporated into the revised manuscript. No additional citation requirements were specified by the editor, and we confirm that no further citations are needed at this stage.

2. Regarding the completeness and correctness of the reference list

We have thoroughly reviewed the entire reference list to ensure:

All cited works are accurate and complete.

No retracted papers are included in the reference list.

All references follow the PLOS ONE format consistently.

Response to Additional Editor Comments:

Editor's Comment: "Thank you for submitting the revised manuscript. Although the major concerns are addressed, some minor corrections are still required."

Our Response: Thank you for this acknowledgment. We have carefully addressed all minor corrections.

Response to Reviewer Comments:

Reviewer #1:

Comment 1: "All comments have been addressed."

Our Response: Thank you for your positive assessment and for your valuable guidance throughout the review process.

Comment 2: "The revisions are accepted, please update the manuscript as it has some typos and needs proofreading."

Our Response: Thank you for this important observation. We have conducted a thorough language proofreading of the entire manuscript to eliminate any typographical or grammatical errors. Specifically:

The manuscript has been carefully reviewed by all co-authors.

We have carefully edited to ensure the manuscript is clear, correct, and unambiguous.

All identified typos and minor language issues have been corrected. Revised portion are marked in red in the manuscript.

The language-polished version of the manuscript is now submitted. We believe it meets the high standards required for publication.

Reviewer #2:

Comment 1: "All comments have been addressed."

Our Response: Thank you for your confirmation and for your constructive suggestions throughout the revision process.

Comment 2 (Questions 2-6): The reviewer responded "Yes" to questions regarding technical soundness, statistical analysis, data availability, and language presentation.

Our Response: We sincerely appreciate the reviewer's positive evaluation of our work.

We thank you once again for your thorough and constructive handling of our manuscript. We believe that with these final corrections, the manuscript is now suitable for publication in PLOS ONE. Should any further adjustments be required, we remain at your disposal.

Yours sincerely,

Xingbo Long, M.D.

Deyang People's Hospital, Deyang, 618000, China

Tel.: +86-18280226483

E-mail: 723766496@qq.com

---

## [Decision Letter · Decision Letter 2]

4 Mar 2026

An empirical study of drone medical logistics transportation in a multi-campus model of Chinese public hospitals: real-world data-driven validation of timeliness and application effects

PONE-D-25-54130R2

Dear Dr. Long,

We’re pleased to inform you that your manuscript has been judged scientifically suitable for publication and will be formally accepted for publication once it meets all outstanding technical requirements.

Kind regards,

Zeashan Hameed Khan, Ph.D.

Academic Editor

PLOS One

Additional Editor Comments (optional):

The revised version has been evaluated for correctness and compliance as per the reviewer's comments and it can be accepted in the present form.

Reviewers' comments:

Reviewer's Responses to Questions

**Comments to the Author**

1. If the authors have adequately addressed your comments raised in a previous round of review and you feel that this manuscript is now acceptable for publication, you may indicate that here to bypass the “Comments to the Author” section, enter your conflict of interest statement in the “Confidential to Editor” section, and submit your "Accept" recommendation.

Reviewer #1: All comments have been addressed

Reviewer #2: All comments have been addressed

2. Is the manuscript technically sound, and do the data support the conclusions?

Reviewer #1: (No Response)

Reviewer #2: Yes

3. Has the statistical analysis been performed appropriately and rigorously? 

Reviewer #1: (No Response)

Reviewer #2: Yes

4. Have the authors made all data underlying the findings in their manuscript fully available?

Reviewer #1: (No Response)

Reviewer #2: Yes

5. Is the manuscript presented in an intelligible fashion and written in standard English?

Reviewer #1: (No Response)

Reviewer #2: Yes

6. Review Comments to the Author

Reviewer #1: (No Response)

Reviewer #2: authors respond all my concerns and can be published

authors respond all my concerns and can be published

7. PLOS authors have the option to publish the peer review history of their article (what does this mean?). If published, this will include your full peer review and any attached files.

Reviewer #1: No

Reviewer #2: **Yes:** Ata Jahangir Moshayedi

---

## [Editor Report · Acceptance letter]

PONE-D-25-54130R2

PLOS One

Dear Dr. Long,

I'm pleased to inform you that your manuscript has been deemed suitable for publication in PLOS One. Congratulations! Your manuscript is now being handed over to our production team.

Kind regards,

on behalf of

Dr. Zeashan Hameed Khan

Academic Editor

PLOS One